# ALS-Associated SOD1(G93A) Decreases SERCA Pump Levels and Increases Store-Operated Ca^2+^ Entry in Primary Spinal Cord Astrocytes from a Transgenic Mouse Model

**DOI:** 10.3390/ijms20205151

**Published:** 2019-10-17

**Authors:** Rosa Pia Norante, Caterina Peggion, Daniela Rossi, Francesca Martorana, Agnese De Mario, Annamaria Lia, Maria Lina Massimino, Alessandro Bertoli

**Affiliations:** 1Department of Biomedical Sciences, University of Padova, 35131 Padova, Italy; rosa.norante@gmail.com (R.P.N.); catpeg@outlook.it (C.P.); agnese.demario@unipd.it (A.D.M.); annamaria.lia@unipd.it (A.L.); 2Laboratory for Research on Neurodegenerative Disorders, Istituti Clinici Scientifici Maugeri SpA SB–IRCCS, 27100 Pavia, Italy; daniela.rossi@icsmaugeri.it (D.R.); francesca.martorana@icsmaugeri.it (F.M.); 3CNR–Neuroscience Institute, University of Padova, 35131 Padova, Italy; marilina.massimino@gmail.com; 4Padova Neuroscience Center, University of Padova, 35131 Padova, Italy

**Keywords:** amyotrophic lateral sclerosis, astrocytes, Ca^2+^ signaling, store-operated Ca^2+^ entry, endoplasmic reticulum, SERCA, transgenic mice, STIM, Orai

## Abstract

Amyotrophic lateral sclerosis (ALS) is a fatal neurodegenerative disorder characterized by the selective death of motor neurons (MNs), probably by a combination of cell- and non-cell-autonomous processes. The past decades have brought many important insights into the role of astrocytes in nervous system function and disease, including the implication in ALS pathogenesis possibly through the impairment of Ca^2+^-dependent astrocyte-MN cross-talk. In this respect, it has been recently proposed that altered astrocytic store-operated Ca^2+^ entry (SOCE) may underlie aberrant gliotransmitter release and astrocyte-mediated neurotoxicity in ALS. These observations prompted us to a thorough investigation of SOCE in primary astrocytes from the spinal cord of the SOD1(G93A) ALS mouse model in comparison with the SOD1(WT)-expressing controls. To this purpose, we employed, for the first time in the field, genetically-encoded Ca^2+^ indicators, allowing the direct assessment of Ca^2+^ fluctuations in different cell domains. We found increased SOCE, associated with decreased expression of the sarco-endoplasmic reticulum Ca^2+^-ATPase and lower ER resting Ca^2+^ concentration in SOD1(G93A) astrocytes compared to control cells. Such findings add novel insights into the involvement of astrocytes in ALS MN damage.

## 1. Introduction

Amyotrophic lateral sclerosis (ALS) is a fatal neurodegenerative disease characterized by the selective damage and death of motor neurons (MNs) in the motor cortex, brain stem and spinal cord [1,2,3,4]. Most of ALS cases (90%) are sporadic, but the disease also occurs on inheritable grounds (familial ALS, fALS), with the majority of fALS associated with mutations in the C9orf72, TDP-43 (TAR DNA-binding protein 43), RNA-binding protein FUS/TLS (Fused in Sarcoma/Translocated in Sarcoma) or superoxide dismutase 1 (SOD1) genes [3]. Studies of the inherited forms have provided much of the current knowledge on ALS pathogenesis by means of cell models as well as transgenic (Tg) animals, including Tg mice expressing the human (h) SOD1(G93A) missense mutation, the most exploited animal model of ALS. Although the exact mechanisms underlying disease onset still remain unclear, there is increasing evidence that both intrinsic pathways and non-cell autonomous events triggered by other cell types, particularly the neighboring astrocytes, concur to MN degeneration [5,6,7]. In this context, it is worth noting that astrocytes, the most abundant non-neuronal cells surrounding MNs, have a pivotal role in the maintenance of neuronal metabolism, e.g., by releasing gliotransmitters that are relevant for astrocyte-to-neuron communication, and sequestering glutamate from the synaptic cleft to protect neurons from glutamate excitotoxicity [8]. Consistently, many pieces of evidence indicate that altered astrocytic signaling concurs to disrupt MN functions and viability in different ALS models. For example, silencing mutant SOD1 expression in astrocytes slows down disease progression in fALS Tg mice in vivo [6,7], while the specific expression of mutant SOD1 and TDP-43 in astrocytes disrupts MN mitochondrial functions in primary astrocyte/MN co-culture models [9], and causes MN death in Tg rats [10], respectively. Accordingly, several other in vivo and in vitro studies reveal many pathogenic changes in ALS astrocytes, including disrupted G_q_-protein-coupled receptor-mediated Ca^2+^ signaling and mitochondrial functional deficiencies [11,12,13], and suggest that altered astrocytic functions and/or release of different (neurotoxic) molecules by the astrocytes may play a major role in MN pathology [6,7,9,11,12,13,14,15,16,17].

Astrocytes are electrically silent, but rely on Ca^2+^ signals, such as local [Ca^2+^] oscillations or Ca^2+^ waves for exerting their (patho)-physiologic functions, including secretion of neurotrophic/neurotoxic factors [18,19,20,21]. In line with this notion, a previous work by Manfredi and colleagues demonstrated that increased (basal) store-operated Ca^2+^ entry (SOCE) in murine hSOD1(G93A)-expressing astrocytes initiated a cascade of cellular alterations, leading to aberrant SNARE-dependent exocytosis and consequent astrocyte-mediated toxicity to MNs [13].

Originally identified in a subset of non-excitable cells, such as immune cells, SOCE has been subsequently characterized in many cell types, including skeletal myocytes, neurons and astrocytes, and has become increasingly recognized as an important mechanism of cell Ca^2+^ entry, distinguished from that mediated by canonical voltage- or ligand-gated channels [22,23,24,25]. SOCE occurs upon depletion of intracellular Ca^2+^ stores (in particular from the endoplasmic reticulum, ER) that leads to the opening of store-operated Ca^2+^ channels located at the plasma membrane (PM), allowing external Ca^2+^ entry in the cells and the replenishment of ER Ca^2+^ stores.

From a molecular point of view, the mechanisms of SOCE activation involve the redistribution into punctate clusters of ER membrane proteins (stromal interaction molecules, STIM1 and 2)–containing luminal Ca^2+^ sensing domains–in response to ER [Ca^2+^] reduction. Such a clustering causes the recruitment and activation of PM pore-forming proteins (Orai1-3;) [22,26,27]. From a functional point of view, not only SOCE controls the overall Ca^2+^ homeostasis by ensuring the proper refilling of intracellular Ca^2+^ stores, but it also concurs to the regulation of several cell functions, including programmed cell death [28], lymphocyte maturation [29], skeletal myocyte differentiation [30], neuronal excitability/signaling and gene regulation [31,32]. In line with this broad range of functions, defective SOCE is implicated in different diseases [26], such as severe combined immunodeficiency [33,34], cancer [35], and neurodegenerative disorders [32].

In light of the above notions, we set out to accomplish a thorough analysis of SOCE in primary astrocytes from the spinal cord of newborn Tg mice expressing the ALS-related hSOD1(G93A) or the non-pathogenic wild-type (WT) hSOD1 proteins. To this purpose, we took advantage of genetically-encoded Ca^2+^ indicators (GECI) targeted to different cell compartments for the recording of local Ca^2+^ responses to SOCE stimulation. Our results confirmed that hSOD1(G93A) astrocytes suffer from aberrant SOCE and underscored lower ER basal Ca^2+^ levels that are associated with reduced expression of the sarco-endoplasmic reticulum Ca^2+^-ATPase SERCA2 in these cells compared to the WT counterpart. Conversely, no difference was observed in SOCE-mediated mitochondrial Ca^2+^ response, and other parameters related to mitochondrial Ca^2+^ homeostasis.

## 2. Results

### 2.1. hSOD1(G93A)-Expressing Astrocytes Have Enhanced SOCE-Mediated Ca^2+^ Influx

Considering the well-recognized role of astrocytes in ALS pathogenesis, and that the control of Ca^2+^ homeostasis is fundamental for proper astrocyte functions [18,19,20,21], we undertook a comparative study of Ca^2+^ dynamics in primary spinal astrocytes from newborn hSOD1(G93A) and hSOD1(WT) mice. In light of a previous study showing increased SOCE in murine hSOD1(G93A) [13], here we thoroughly analyzed Ca^2+^ fluxes in different cell compartments/domains of live cells using different GECIs, in order to define a more precise picture of SOCE dysregulation in ALS astrocytes.

Firstly, intracellular Ca^2+^ dynamics following SOCE activation were evaluated by means of the bioluminescent photo-protein aequorin (AEQ), targeted to cytosolic PM micro-domains (AEQ_PM_) or the bulk cytosol (AEQ_Cyt_). The transient rise of both [Ca^2+^]_PM_ (Figure 1A) and [Ca^2+^]_Cyt_ (Figure 1B) had a significantly higher (of 30% and 25%, respectively) peak value in hSOD1(G93A) than in hSOD1(WT) astrocytes, indicating that hSOD1(G93A) expression leads to higher SOCE-mediated Ca^2+^ replenishment of astrocytes.

### 2.2. SOCE-Mediating Proteins Are Equally Present in hSOD1(WT) and hSOD1(G93A) Primary Astrocytes

The mechanism of SOCE activation involves two families of ER (STIM1 and 2) and PM (Orai1-3) proteins [22,36,37]. STIM1 and STIM2 sense luminal Ca^2+^ changes with different affinity for the ion [38], thereby recruiting and activating the PM channel-forming Orai proteins and promoting pore opening [37,39,40].

In light of the results of Figure 1, we asked whether enhanced SOCE in hSOD1(G93A) astrocytes was attributable to increased expression of the above molecular components of the SOCE mechanism, i.e., STIM1, STIM2, Orai1 and Orai3 (that are the Orai family members most abundant in astroglial cells [41,42]). Western blot (WB) analyses, however, resulted in no significant difference in the expression of such proteins in primary astrocytes bearing the two hSOD1 forms (Figure 2).

### 2.3. hSOD1(G93A) Astrocytes Have Reduced basal Ca^2+^ Levels in the Cytosol and the ER Lumen Compared to Control Astrocytes

Considering that the local cell Ca^2+^ homeostasis results from the fine regulation of several mechanisms, we next evaluated whether differences in the resting (basal) Ca^2+^ levels in different cellular compartments could account for the increased SOCE observed in hSOD1(G93A) astrocytes.

To this purpose, we performed Ca^2+^ imaging in hSOD1(WT) and hSOD1(G93A) primary astrocytes by means of different Ca^2+^ indicators, including fluorescent GECIs (fluorescence resonance energy transfer (FRET)-based cameleons, and GEM-Cepia1ER) and the chemical dye Fura-2 (all of which are suitable for single-cell Ca^2+^ measurements). For GECI-based analyses, cells were transfected with expression plasmids encoding the Ca^2+^ probes targeted to the cytosol or the ER lumen. Basal [Ca^2+^] were recorded in 2 mM external [Ca^2+^] using suitable computer-assisted fluorescence microscopy workstations. Measurements with either the cytosolic-targeted cameleon (D1cpv) or Fura-2 showed that hSOD1(G93A) astrocytes have significantly reduced basal cytosolic [Ca^2+^] levels compared to non-ALS controls (Figure 3A,B, respectively). Similarly, both the ER-targeted GECIs, D4ER cameleon and GEM-Cepia1ER, indicated lower luminal ER Ca^2+^ levels in hSOD1(G93A) astrocytes under resting conditions (Figure 3C,D, respectively). Taken together, these results indicate that hSOD1(G93A)-expressing astrocytes have lower Ca^2+^ levels at resting conditions both in the cytosol and the ER. Importantly, the lower basal [Ca^2+^] in the ER lumen may contribute to render hSOD1(G93A) astrocytes more sensitive to SOCE activation and cause alterations in other ER-dependent cellular processes (see below).

### 2.4. SERCA2 Is Down-regulated in hSOD1(G93A) Astrocytes

The maintenance of proper cell Ca^2+^ concentrations is guaranteed by an efficient network of Ca^2+^-transporting pathways and by Ca^2+^-buffering proteins present in the cytosol and in the lumen of Ca^2+^-storing compartments [43]. The ER is generally considered as the major and most metabolically relevant reservoir of, and buffering system for, intracellular Ca^2+^. The ER lumen has high Ca^2+^ concentration and contains several Ca^2+^-binding proteins [44]. Ca^2+^ uptake in, and release from, the ER affect many processes, not only in the lumen of the organelle, but also in other cell compartments. For example, changes in cytosolic and mitochondrial Ca^2+^ levels consequent to ER Ca^2+^ handling can, in turn, affect numerous signal transduction pathways [45,46]. Considering that the control of ER Ca^2+^ metabolism is key to cell Ca^2+^ homeostasis, and that perturbed ER Ca^2+^ has been proposed as a possible mechanism in ALS pathogenesis [47,48,49], we analyzed by WB the expression of ER Ca^2+^-binding proteins or Ca^2+^-transporting systems.

We found that Ca^2+^-binding proteins, such as calnexin (CLNX) and calreticulin (CRT), and the unfolded protein response regulator Grp78/BiP are equally expressed in astrocytes of the two different hSOD1 genotypes (Figure 4A–C,E). Conversely, our data showed that the sarco-endoplasmic reticulum Ca^2+^ ATPase (SERCA) isoform 2 (SERCA2, the SERCA isoform present in the nervous system [50]), which is one of the major systems involved in [Ca^2+^] control by pumping calcium ions into the ER, is downregulated in hSOD1(G93A) astrocytes compared to the hSOD1(WT) counterpart (Figure 4D,E). SERCA2 downregulation is likely to occur by a post-transcriptional or post-translational effect, since mRNA quantification of SERCA2b (the SERCA2 splicing variant more expressed in the nervous system) by real-time RT-PCR showed no statistically significant difference between the two hSOD1 types (Appendix A). This effect seems to be specific for SERCA, because the two major PM systems involved in cytosolic Ca^2+^ clearance, i.e., the PM Ca^2+^ ATPase (PMCA) and the Na^+^/Ca^2+^ exchanger (NCX), were equally expressed in fALS-related hSOD1(G93A) and healthy astrocytes (Appendix A).

### 2.5. Mitochondria of hSOD1(G93A) and hSOD1(WT) Astrocytes Equally Respond to SOCE Stimulation

In addition to the ER, it is nowadays largely accepted that also mitochondria play a primary role in cell Ca^2+^ buffering [51,52] by actively taking up the ion in the mitochondrial matrix through the mitochondrial Ca^2+^ uniporter (MCU) complex located in the inner mitochondrial membrane [52,53,54], and thanks to the sustained mitochondrial membrane potential (Δψ_m_). Furthermore, Ca^2+^ ions play a fundamental role in several mitochondrial functions [52], and (Ca^2+^-related) mitochondrial defects (also in astrocytes) were repeatedly correlated to ALS pathogenesis [9,11,55,56]. We thus analyzed the Ca^2+^ uptake capability of mitochondria upon SOCE stimulation in primary spinal cord astrocytes by means of a mitochondrially-targeted AEQ (AEQ_mit_). Quite surprisingly, we found no difference in SOCE-induced mitochondrial Ca^2+^ transients between healthy and ALS astrocytes (Figure 5), suggesting that the mitochondrial Ca^2+^ buffering capacity plays no significant role in the excess of cytosolic Ca^2+^ accumulation observed in hSOD1(G93A) after SOCE stimulation.

### 2.6. Ca^2+^-related Mitochondrial Parameters Are Comparable in hSOD1(G93A) and hSOD1(WT) Astrocytes

In an attempt to explain the above paradox, we compared an array of (Ca^2+^-related) mitochondrial parameters in astrocytes with the two hSOD1 genotypes. We first analyzed the basal Ca^2+^ levels in mitochondria by means of the 4mtD3cpv cameleon targeted to the mitochondrial matrix, which–however–showed no difference between the two strains (Figure 6A).

Also, the expression of the ion-translocating subunit of the MCU complex did not significantly differ between hSOD1(G93A) and hSOD1(WT) astrocytes (Figure 6B). It is worth noting, however, that the composite regulation of MCU activity implicates other proteins of a multi-subunit complex, and thus demands further studies.

Because the mitochondrial Δψ_m_ is the driving force for mitochondrial Ca^2+^ uptake, we subsequently compared Δψ_m_ in the two astrocyte types using the potentiometric probe tetramethylrhodamine methyl ester (TMRM). We found no significant variation in Δψ_m_ of hSOD1(G93A) astrocytes compared to the hSOD1(WT) counterpart under basal conditions (Figure 6C).

Finally, we analyzed the mitochondrial morphology by ectopically expressing in astrocytes a mitochondrially-targeted red fluorescent protein (Figure 6D), and evaluating the mitochondria circularity that is an index of mitochondrial elongation [57], which, again, revealed no difference between the two hSOD1 genotypes.

Together, these results indicate that no obvious abnormality occurs in mitochondrial parameters related to the control of Ca^2+^ homeostasis in hSOD1(G93A) astrocytes under basal conditions, arguing against a substantial involvement of mitochondria in the enhanced SOCE observed in these cells.

## 3. Discussion

Given the established *in-trans* effects of astrocytes on MN death in ALS [5,6,7,13,58,59], and the possible involvement of astrocytic Ca^2+^ dysregulation in such a process [11,12,13], here we carried out a comparative analysis of SOCE in primary spinal astrocytes isolated from newborn hSOD1(G93A) and hSOD1(WT) Tg mice by means of locally targeted GECIs.

SOCE is a mechanism of Ca^2+^ transport that is aimed at refilling intracellular Ca^2+^ stores and finely tuning Ca^2+^ homeostasis [22,26], and involves the ER Ca^2+^-sensing STIM1 and 2, and the PM pore-forming Orai1-3, proteins [22,36,37]. STIM1 and STIM2 detect luminal Ca^2+^ changes with different sensitivity [38]. STIM1, whose EF-hand motif binds Ca^2+^ with high affinity (ideal to sense a substantial luminal ER [Ca^2+^] decrease), is uniformly distributed in the ER membrane at resting conditions, whereas it oligomerizes upon ER Ca^2+^ depletion into membrane *punctae* closely juxtaposed to the PM. Such a conformational transition recruits the Orai proteins leading to pore opening, possibly through protein-protein interactions [37,39,40]. Thus, STIM1 is mainly involved in re-establishing resting ER Ca^2+^ levels after the discharge of the ion from the lumen of the organelle. Conversely, STIM2–displaying a lower Ca^2+^ affinity compared to STIM1 [60]–is sensitive to mild reductions of ER Ca^2+^ levels and forms Orai-recruiting *punctae* already at resting ER Ca^2+^ concentrations, thereby tuning the luminal concentration of the ion under non-stimulated conditions [38,61].

SOCE occurs also in astrocytes [42,62], where it contributes to the precise control of Ca^2+^ oscillations that are fundamental for astrocytic signaling and functions. Not surprisingly, SOCE alteration in astrocytes has been reported to be associated with pathological states, such as hypertension and stroke, caused by a loss-of-function mutation in the STIM1 gene in a rat model [63].

In line with the above notions, the work by Manfredi and colleagues [13] showed that oxidative stress-induced *S*-glutathionylation and clustering of the ER Ca^2+^ sensor STIM1 determine abnormally high SOCE in murine hSOD1(G93A)-expressing astrocytes. Such occurrence seemed to happen in a Ca^2+^-store-independent fashion also under basal conditions (i.e., in the absence of specific SOCE stimulation) [64]. The consequent ER Ca^2+^ overload, would result in excessive Ca^2+^ release from the ER upon stimulation of metabotropic purinergic receptors, which would, in turn, induce excitotoxic (to MNs) ATP release from hSOD1(G93A) astrocytes.

In this work, we undertook a different approach, aimed at observing Ca^2+^ fluctuations in different cell compartments upon specific SOCE activation. Our GECI-based study confirmed most of the previous findings, definitively demonstrating higher SOCE in hSOD1(G93A) astrocytes compared to the healthy counterpart. We also reported that increased SOCE in hSOD1(G93A) astrocytes does not depend on altered expression levels of the STIM or the Orai protein family members. On the other hand, in line with previous observations [65], we detected an enhancement in total protein *S*-glutathionylation in hSOD1(G93A) cells (Appendix A), supporting the contention that increased SOCE could be due, at least in part (see below), to the *S*-glutathionylation-dependent formation of STIM1-containing *punctae* at the PM under resting conditions [13].

An important unprecedented observation of our work is the finding that the SERCA2 pump is remarkably downregulated (by around 40%) in hSOD1(G93A) astrocytes compared to hSOD1(WT). This effect seems to be specific for SERCA, since we found no significant difference in the levels of other ER (Ca^2+^-related) proteins (i.e., CLNX, CRT, and Grp78), and PM Ca^2+^-transporting systems that are crucially involved in the control of cell Ca^2+^ homeostasis (i.e., PMCA and NCX). In addition, quantitative RT-PCR indicated that SERCA2 downregulation likely occurs by (still unknown) post-transcriptional or post-translational mechanisms. Further studies are, however, required to better address this issue.

Reduced SERCA levels could explain the slight, yet significant, reduction of resting ER Ca^2+^ content of hSOD1(G93A) astrocytes that we reported here by using two different ER-targeted GEGIs (i.e., the D4ER cameleon and GEM-Cepia1ER). This finding suggests that the ER of hSOD1(G93A) astrocytes has a lower capacity to drain cytosolic Ca^2+^ through the pumping activity of SERCA, and that Ca^2+^ store-dependent processes (i.e., a prompter activation of SOCE by reduced basal ER Ca^2+^-levels) might concur to generate the higher SOCE observed in these cells, compared to the hSOD1(WT) counterpart. It is also important to underline that a chronically reduced Ca^2+^ concentration in the ER lumen could also trigger defects in a series of ER-dependent functions, including ER-mediated protein control, that have been often associated to ALS pathogenesis [48,49,66].

The above observation is seemingly in contrast with the previous report of higher ER basal Ca^2+^ stores in hSOD1(G93A) astrocytes [13], a discrepancy that could be rationalized on methodological grounds. For example, we used targeted GECIs, which allows a direct assessment of ER Ca^2+^ homeostasis (and basal concentration), rather than monitoring–as an indirect index–the stimulated Ca^2+^ release from the ER, using cytosolic Ca^2+^ indicators. In addition, it is renown in the literature that many astrocyte patho-physiological features, including Ca^2+^ control, may differ between newborn and adult animals and/or may be conditioned by the central nervous system region they belong to [67,68,69]. For example, hSOD1(G93A) induces early (pre-symptomatic) alterations only in a MN-associated subset of astrocytes in Tg mice [70,71]. With respect to this, it is worth noting that our analyses of ER Ca^2+^ stores were carried out on primary astrocytes isolated from the spinal cord of newborn mice. This is at variance with the previous study by Kawamata and colleagues [13], who used primary spinal astrocytes from adult mice or cortical astrocytes from newborns. Since newborn hSOD1(G93A) mice display very limited motor neuron death, astrocyte activation, and inflammation and do not show any clinical symptom of the disease, one can conceive that astrocytic cultures from aged animals would have highlighted more severe SOCE-related phenotypes. On the other hand, however, our choice of using such an experimental paradigm discloses the possibility that early modifications of Ca^2+^-dependent astrocyte-to-neuron communication play an etiological role in ALS.

Differently from the more pronounced Ca^2+^ transients recorded both in cytosolic sub-PM microdomains and the bulk cytosol following SOCE stimulation, we observed (by use of two different probes) that under resting conditions a lower cytosolic Ca^2+^ concentration pertains to hSOD1(G93A) astrocytes. Such a finding seemingly disagrees with the reduced SERCA expression and ER lower Ca^2+^-uptake capacity of these cells, which would more readily account for higher basal cytosolic [Ca^2+^]. It is worth noting, however, that basal cell Ca^2+^ levels are tightly controlled by a complex array of Ca^2+^-transporting/buffering systems. The precise mechanisms leading to such a chronic unbalance of cell Ca^2+^ homeostasis in hSOD1(G93A) astrocytes, which may weaken ALS astrocytes themselves and contribute to their long-term degeneration [12], as well as perturb (Ca^2+^-dependent) astrocyte-MN cross-talk [5,11], remain to be determined and deserve further investigation.

We also directly monitored mitochondrial Ca^2+^ transients in stimulated ALS astrocytes by means of a specific GECI targeted to the mitochondrial matrix. With respect to this, an unexpected result was the observation that, in spite of the higher SOCE-dependent Ca^2+^ rise in the cytosol of hSOD1(G93A)-expressing astrocytes, we did not detect an increase in Ca^2+^ transients in the mitochondrial matrix. Such a finding is in line with a previous report [13], though this result remains puzzling in view of the reduced Ca^2+^ buffering capability of the ER, and considering that other Ca^2+^-extruding systems (i.e., PMCA and NCX), as well as the measured mitochondrial Ca^2+^ parameters, are unchanged in hSOD1(G93A) cells compared to the WT counterpart. Since mitochondria substantially contribute to Ca^2+^ uptake from, and consequent Ca^2+^ buffering in, the cytosol [51,52], this issue deserves further investigation (e.g., expression profiling of other subunits of the MCU complex, such as MICU1-2, EMRE).

In the context of Ca^2+^ dysregulation in hSOD1(G93A) astrocytes, previous work by co-authors of the present paper highlighted aberrant Ca^2+^ signaling in primary astrocytes from the spinal cord of newborn hSOD1(G93A) mice following stimulation of group I metabotropic glutamate receptors (mGluRs) [12]. Such a compromised Ca^2+^ regulation involved the disruption of Ca^2+^ oscillations, which play a crucial role in both astrocyte-neuron cross-talk and maintenance of astrocyte survival under normal circumstances. This condition was linked to uncontrolled cytosolic Ca^2+^ overload due to massive release of the ion from the ER through inositol trisphosphate receptor channels. Such events correlated with cytochrome c release from mitochondria that activated downstream cell death pathways. Using the same cell type of the previous report, here we further investigated the Ca^2+^ signals occurring in astrocytes expressing the ALS-linked hSOD1(G93A) mutant. We report an increase in cytosolic Ca^2+^ also upon a different stimulus, i.e., SOCE. Interestingly, we find that SOCE-induced Ca^2+^ transients are coupled to unvaried mitochondrial Ca^2+^ uptake, Δψ_m_ and morphology, irrespective of the presence of the mutant hSOD1 transgene. The preservation of these mitochondrial parameters can be explained by the different time-scale of the two used stimuli (SOCE-induced Ca^2+^ transients in the present work; sustained mGluR activation in the preceding work) as well as by the duration of the downstream Ca^2+^ responses (Ca^2+^ transients lasting tens of seconds vs. prolonged registrations in the range of minutes).

In spite of overwhelming evidence, a unifying explanation of the in-trans implication of astrocytic derangements in ALS-related MN demise is still missing. Here we added a further tile to such a complex mosaic by reporting an unprecedented remarkable reduction in SERCA2 expression in primary cultured astrocytes from the spinal cord of newborn Tg mice expressing the ALS-associated hSOD1(G93A) mutant compared to hSOD1(WT) counterpart. Such a finding is consistent with reduced basal ER Ca^2+^ levels, which may in turn contribute to higher Ca^2+^ transients following SOCE stimulation. Therefore, in addition to the store-independent basal SOCE upregulation (associated to oxidative stress-dependent STIM1 *S*-glutathionylation) [13], also a store-dependent effect may contribute to increased SOCE upon specific stimulation conditions. Under the same settings, mitochondria seem not to contribute substantially to the observed SOCE-related Ca^2+^ phenotype.

In conclusion, it is sensible to envisage that the reduced Ca^2+^ buffering capability of the ER in hSOD1(G93A)-expressing astrocytes compromises the proper response of this cell type to Ca^2+^-mobilizing cues (e.g., SOCE (present data), glutamate [12] or ATP [13]), thereby resulting in abnormally high intracellular Ca^2+^ variations that possibly harm MNs through the activation of different Ca^2+^-dependent neurotoxic pathways.

## 4. Materials and Methods

### 4.1. Mouse Models and Breeding

hSOD1(WT) and hSOD1(G93A) Tg mice (B6SJL(Tg-SOD1)2Gur/J and B6SJL(Tg-SOD1*G93A)1Gur/J mice, respectively) were purchased from The Jackson Laboratory (Bar Harbor, ME, USA, cat. n. 002297 and 002726, respectively). These mice overexpress human WT or mutant (G93A) SOD1 mimicking the natural pattern of SOD1 expression, or the human fALS phenotype, respectively [72,73]. The hSOD1(WT) B6SJL(Tg-SOD1)2Gur/J Tg strain, purposedly generated to serve as a control for the B6SJL(Tg-SOD1*G93A)1Gur/J strain and originally published as N1029 [72], has been reported to express the same protein levels of the SOD1 transgene as the hSOD1(G93A) B6SJL(Tg-SOD1*G93A)1Gur/J Tg mice, even though the transgene copy number in the hSOD1(G93A) strain (~20 copies) is higher.

The colonies were maintained by breeding hemizygous Tg males to non-Tg B6SJLF1/J hybrid females. Embryos and newborns were genotyped following standard procedures as previously described [74].

All aspects of animal care and experimentation were performed in compliance with European and Italian (D.L. 26/2014) laws concerning the care and use of laboratory animals. The Authors’ Institution has been accredited for the use of experimental mice by the Italian Ministry of Health (Authorization N. 305/2017-PR, released on 6 April 2017), and by the ethical committee of the University of Padova (*Organismo Preposto al Benessere degli Animali*, OPBA).

### 4.2. Preparation of Primary Cultures of Murine Spinal Cord Astrocytes

Primary spinal astrocytes were isolated from newborn mice (post-natal day 1–2) and cultured as described in Martorana et al. (2012) [12], with minor modifications. Briefly, after animal sacrifice by decapitation, the spinal cord was explanted and dissected in Hank’s Buffer (Sigma, Saint Louis, MO, USA), and then incubated (5 min, 37 °C) in the same solution added with collagenase (Sigma, 0.25% (*w*/*v*)). After gravity sedimentation, the collagen-free tissue was resuspended in Hank’s Buffer containing DNase I (Roche, Basel, Switzerland, 0.05% (*w*/*v*)), mechanically dissociated by pipetting through a flame-polished glass pipette, added with 2 volumes of minimal essential medium (MEM, Gibco, Thermo Fisher Scientific, Waltham, MA, USA) supplemented with fetal bovine serum 20% (*v*/*v*), and finally centrifuged (220× *g*, 5 min). Cells were resuspended in culture medium (MEM supplemented with fetal bovine serum 20% (*v*/*v*), l-glutamine 4 mM, glucose 0.3% (*w*/*v*), penicillin 100 U/mL, streptomycin 100 μg/mL), and seeded in 25 mL culture flasks. After three weeks of culturing, primary astrocytes reached confluence, were detached from the culture flasks by trypsinization (using a suited Trypsin/EDTA solution, Euroclone, Milano, Italy) and re-plated onto multi-well culture plates or glass coverslips, depending on the experiment, as detailed below. All measurements were carried out 96 h after re-plating. Immuno-cytochemical analyses (see Appendix A) confirmed that, at this stage, cultures contained almost only glial fibrillary acidic protein (GFAP)-positive astrocytes (≈95% over the total cell content, see Appendix A). Cell morphology did not significantly differ between the two hSOD1 genotypes.

### 4.3. Ca^2+^ Imaging

AEQ-based Ca^2+^ measurements. AEQ probes (suited for cell population Ca^2+^ measurements), targeted to cytosolic domains proximal to the plasma membrane (PM) (AEQ_PM_), the bulk cytosol (AEQ_Cyt_) and the mitochondrial matrix (AEQ_mit_) were employed to measure Ca^2+^ fluxes following SOCE stimulation. AEQ-encoding lentiviral vectors were generated, and lentiviral particles purified, as previously described [75,76,77]. Astrocytes, plated onto glass coverslips at the optimal density of 80,000 cells/well in 24-well plates, were infected with lentivectors 24 h after plating and used for experiments 72 h after infection.

AEQ-based Ca^2+^ measurements were performed by means of a computer-assisted luminometer equipped with a perfusion system. For SOCE activation, astrocytes were first incubated (1 h, 37 °C, 5% CO_2_) in Ca^2+^-free Krebs-Ringer buffer ((KRB, NaCl 125 mM, KCl 5 mM, Na_3_PO_4_ 1 mM, MgSO_4_ 1 mM, glucose 5.5 mM, HEPES 20 mM, pH 7.4)), supplemented with EGTA (100 μM), to fully deplete intracellular Ca^2+^ stores, and the AEQ cofactor coelenterazine (5 μM, Santa Cruz Biotechnology, Dallas, TX, USA), in order to reconstitute functional AEQ holoproteins. After transferring the cell-containing coverslip to the thermostatted chamber of the luminometer, experiments started by perfusing cells with the above KRB, first containing EGTA (100 μM), and then CaCl_2_ (1 mM) to maximally stimulate SOCE-mediated Ca^2+^ entry.

Experiments ended by lysing cells with digitonin (100 μM, Sigma) in a hypotonic Ca^2+^-rich solution (CaCl_2_ 10 mM in H_2_O) to discharge the remaining AEQ pool. The light signal was digitalized and stored for subsequent analyses. Luminescence data were calibrated off-line into [Ca^2+^] values, using a computer algorithm based on the Ca^2+^ response curve of AEQ [78].

Cameleon-based Ca^2+^ measurements. FRET-based cameleon GECI were used to measure basal Ca^2+^ levels in the cytosol, the ER lumen and the mitochondrial matrix. Plasmidic vectors coding for the D1cpv and 4mtD3cpv cameleons (targeted to the cytosol and the mitochondrial matrix, respectively) were a kind gift by Roger Tsien (University of California, San Diego, CA, USA; [79,80], while the plasmid for the ER lumen-targeted D4ER was kindly provided by P. Pizzo (Dept. of Biomedical Sciences, University of Padova; [81]. Astrocytes were seeded onto 18 mm-diameter glass coverslips at the optimal density of 160,000 cells/well in 12-well plates and transfected (24 h after plating) with plasmids encoding the desired probe, using the lipofectamine 2000^®^ reagent (Invitrogen, Thermo Fisher Scientific, Waltham, MA, USA) following the manufacturer’s instructions. 72 h after cell transfection, coverslips were mounted into an open-topped chamber and maintained under perfusion with a modified KRB (NaCl 125 mM, KCl 5 mM, KH_2_PO_4_ 0.4 mM, MgSO_4_ 1 mM, CaCl_2_ 2 mM, glucose 5.5 mM, HEPES 20 mM, pH 7.4) through a temperature-controlled (37 °C) instrument (TC-324B, Warner Instruments), in order to monitor basal Ca^2+^ levels in astrocytes.

FRET signals were acquired using a computer-assisted DM6000 inverted microscope (Leica Microsystems, Leica, Wetzlar, Germany) with a 40× oil objective (HCX Plan Apo, NA 1.25), coupled to an IM 1.4C cooled CCD (Jenoptik Optical Systems, Jena, Germany) attached to a 12-bit frame grabber. Excitation light produced by a 410 nm led (LZ1-00UA00-LED, Led Engin, Osram, Munich, Germany) was filtered at the appropriate wavelength (425 nm) through a band-pass filter, and the emitted light was collected through a dichroic mirror (515 DCXR, Chroma Technology, Bellow Falls, VT, USA), and a beam-splitter (OES) with emission filters (Chroma Technology) HQ 480/40M (for CFP) and HQ 535/30M (for YFP). The beam-splitter permits the collection of the two emitted wavelengths at the same time, thus preventing any artefact due to uncontrolled movement of the recording chamber and/or intracellular organelles. Images were acquired using an IM 1.4C cooled CCD (Jenoptik Optical Systems) attached to a 12-bit frame grabber. Synchronization of the excitation source and the CCD was performed through a control unit ran by a custom-made software package (developed by C. Ciubotaru, Venetian Institute of Molecular Medicine, Padova, Italy), which was also used for image acquisition, with an exposure time of 100 ms and a time-shift of 0.5 s. The software allows with-time recording of the FRET-acceptor (yellow fluorescent protein, YFP) and -donor (cyan fluorescent protein, CFP) fluorescence intensity (sampling rate = 1 s^−1^). FRET data were then off-line analyzed using the ImageJ software for the calculation of the ratio between the YFP (535 nm) and CFP (480 nm) emission signals, which is taken as a relative measurement of Ca^2+^ concentration. Before such a calculation, the emission fluorescence intensity of YFP and CFP in proper regions of interest (ROIs) on selected cells was subtracted for the respective background signal. Representative fluorescence micrographs of astrocytes (under resting conditions) expressing the different cameleon probes for both the YFP and the CFP channels are reported in Appendix A.

Fura-2-based cytosolic Ca^2+^ measurements. For Fura-2 analyses, astrocytes were seeded onto 18 mm-diameter cover slips at the optimal density of 160,000 cells/well in 12-well plates. 96 h after plating, cells were incubated (20 min, 37 °C) with Fura-2/AM (1 μM), pluronic F-127 (0.02% (*w*/*v*)), and sulfinpyrazone (200 μM) in a modified KRB (NaCl 140 mM, KCl 2.8 mM, MgCl_2_ 1 mM, CaCl_2_ 2 mM, glucose 11 mM, HEPES 10 mM, pH 7.4), and then left 20 min at room temperature in the above modified KRB to allow de-acetylation of the probe. Fura-2-loaded cells were then processed for Ca^2+^ imaging using an inverted microscope (Axiovert 100, Zeiss, Oberkochen, Germany) equipped with a 40× ultraviolet-permeable objective (Olympus Biosystems, Hamburg, Germany). Alternating excitation wavelengths of 340 and 380 nm were obtained by a monochromator controlled by a custom software (developed by C. Ciubotaru, Venetian Institute of Molecular Medicine, Padova, Italy). A neutral density UVND 0.6 filter (Chroma Technology) was used in the excitation pathway. The emitted fluorescence was measured at 500–530 nm. Images were acquired every 5 seconds, with a 200 ms-exposure time at each wavelength, by a SensiCam QE camera (PCO AG, Kelheim, Germany) controlled by the same software. ROIs corresponding to the entire cell soma were selected for Ca^2+^ imaging, and the ratio of the emitted fluorescence intensities (F340/F380) was averaged offline after background subtraction.

GEM-Cepia1ER-based Ca^2+^ measurements. To monitor basal ER Ca^2+^ levels by means of the GEM-Cepia1ER GECI [82], astrocytes were plated onto 18 mm-diameter glass coverslips at the optimal density of 160,000 cells/well in 12-well plates, and transfected (24 h after plating) with the probe-encoding plasmid (kindly provided by. P. Pizzo, Dept. of Biomedical Sciences, University of Padova) using the Lipo-2000^®^ reagent (Invitrogen). Cells were analyzed 72 h after transfection by means of an inverted microscope (ECLIPSE Ti, Nikon, Shinagawa, Tokyo, Japan) with a 40× oil-immersion objective (Fluar; NA, 1.30, Zeiss). Ca^2+^ recordings were performed in the same modified KRB used for the cameleon probes (see above). Excitation light (395 nm) was provided by a monochromator (Polychrome V; TILL Photonics, Graefelfing, Germany). The emitted fluorescence was collected at 480 nm and 530 nm for the Ca^2+^-bound and the Ca^2+^-free indicator, respectively, and the ratio of the emitted fluorescence at the two emission wavelengths (480/530), after background subtraction, was taken as a measure of resting ER Ca^2+^ concentration. Images were acquired at 1 Hz, with a 200 ms-exposure time, using a computer-assisted cooled CCD camera (SensiCam QE, PCO AG), and analyzed as described previously [82,83]. Representative fluorescence micrographs of GEM-Cepia1ER-expressing astrocytes (under resting conditions) at both 480 nm and 530 nm emission wavelengths are reported in Appendix A.

### 4.4. Measurement of the Mitochondrial Membrane Potential

The mitochondrial Δψ_m_ of cultured astrocytes was measured using the membrane-permeant TMRM, a cationic red-orange fluorescent dye (λ_ex_ = 548 nm, λ_em_ = 574 nm) that accumulates electrophoretically into the mitochondrial matrix, thereby allowing to correlate the Δψ_m_ amplitude to the fluorescence signal quantification [84]. To this purpose, astrocytes, seeded onto 18 mm-diameter glass coverslips at the optimal density of 160,000 cells/well in 12-well plates and grown for 96 h, were loaded with TMRM (Molecular Probes) by incubation with the probe (10 nM, 30 min, 37 °C, CO_2_ 5%) in CaCl_2_ (1 mM)-containing modified KRB (NaCl 125 mM, KCl 5 mM, KH_2_PO_4_ 1 mM, MgSO_4_ 1 mM, glucose 5.5 mM, HEPES 20 mM, pH 7.4), and then incubated in the above modified KRB devoid of Mg^2+^. Fluorescence images were collected with an inverted microscope (Olympus IMT-2) equipped with a (75W) Xenon lamp, a cooled 16-bit CCD camera (Micromax, Princeton Instruments, Trenton, NJ, USA), a 40× oil-immersion objective, and appropriate excitation and emission filters. Several fields were acquired from each coverslip before and after addition of trifluorocarbonylcyanide phenylhydrazone (FCCP) (5 μM, Sigma) that, by collapsing Δψ_m_, releases the probe from mitochondria. The TMRM fluorescence intensity in each image was quantified in mitochondria-rich ROIs using the Image J software and reported as the difference between the mean fluorescence intensity before and after FCCP addition. Representative fluorescence micrographs of TMRM-loaded astrocytes, before and after FCCP addition, are reported in Appendix A.

### 4.5. Analysis of Mitochondrial Morphology

Primary astrocytes, seeded onto 13 mm-diameter glass coverslips at the optimal density of 80,000 cells/well in 24-well plates, were transiently transfected (24 h after plating) using the Lipofectamine 3000^®^ reagent (Invitrogen), according to the manufacturer’s instructions, with a plasmid coding for mtRFP (mitochondrially-targeted red-fluorescent protein). 72 h after transfection, cells were observed by means of a laser scanning confocal microscope TCS-SP5 (Leica Microsystems) equipped with a 40× oil-immersion objective to evaluate mitochondrial morphology. To this purpose, fluorescence microscopy images of randomly selected fields were acquired (at least 10 fields for each coverslip), and every single detected mitochondrion was analyzed for its morphological characteristics, such as surface area, and major and minor axes by means of the Volocity software (Improvision, Perkin-Elmer, Waltham, MA, USA). On the basis of such parameters, the mitochondrion circularity shape factor (4π × area/perimeter^2^) was calculated as an inverse index of mitochondrial elongation [57,85], approximating mitochondria to elliptical structures.

### 4.6. Western Blot Analysis

96 h after plating onto 12-well plates (at a density of 160,000 cells/well), astrocytes were homogenized in a lysis buffer containing glycerol 10% (*w*/*v*), sodium dodecyl sulfate (SDS) 2% (*w*/*v*), Tris 62.5 mM (pH 6.8), NaVO_4_ 5 mM, protease and phosphatase inhibitor cocktails (Roche), and cell lysates were boiled (5 min) to accomplish complete protein denaturation. The total protein content was determined by the Lowry method (Total Protein Kit, Micro Lowry, Peterson’s Modification, Sigma), using BSA as standard. The reducing agent dithiothreitol (50 mM) and bromophenol-blue (0.004% (*w*/*v*)) were added to samples just before gel loading. SDS-polyacrylamide gel electrophoresis (SDS-PAGE) was performed on precast gels (4–15% acrylamide/bis-acrylamide gradient concentration, Bio-Rad, Hercules, CA, USA), loaded with 18 μg of proteins per lane and run in a Tris/glycine buffer. Proteins were then electro-blotted onto polyvinylidene difluoride (PVDF) membranes (0.22 μm pore size, Bio-Rad). Membranes were incubated (1 h, RT) with a blocking solution of Tris-buffered saline added with Tween-20 (0.1% (*w*/*v*)) (TBS-T), containing non-fat dry milk 5% (*w*/*v*), or BSA 3% (*w*/*v*), and then incubated (over-night, 4 °C) in the same blocking solution containing the desired primary antibody (Ab) (see below). Finally, after three 10 min-washes (with TBS-T), membranes were incubated (1 h, RT) with a horseradish peroxidase-conjugated anti-rabbit or anti-goat IgG secondary Ab (Santa Cruz Biotechnology), depending on the used primary Ab. After washing in TBS-T, immunoreactive bands were visualized and digitalized by means of a NineAlliance digital camera workstation (UVITEC), using an enhanced chemiluminescence reagent kit (Millipore, Burlington, MA, USA). For densitometric analyses, the optical density of the immunoreactive bands of interest was normalized to the optical density of the corresponding lane in the PVDF membrane stained with Coomassie blue [86]. Although in the figures only a portion of the Coomassie-stained PVDF membrane was shown for the sake of space, the entire lanes shown in Appendix A were used for densitometric analyses.

Antibodies. The following polyclonal (p) Abs were used (dilution in parentheses): anti-STIM1 (1:1000; Cell Signaling Technology, Danvers, MA, USA, cat. n 4916); anti-STIM2 (1:1000; ProSci, Poway, CA, USA, cat. n. 4123); anti-Orai1 (1:1000; Abcam, Cambridge, UK, cat n. ab59330); anti-Orai3 (1:1000; Sigma, cat. n. SAB3500447); anti-SERCA2 (1:500; Santa Cruz Biotechnology, cat. n. sc-8094); anti-MCU (1:1000; Sigma, cat. n. HPA016480); anti-CRT (1:500; Stressgen, San Diego, CA, USA, cat. n. ADI-SPA-600-J), anti-CLNX (1.1000; Stressgen, cat. n. ADI-SPA-865), anti-Grp78 (1:1000; Abcam, cat. n. 3183). All pAbs were raised in rabbit, with the exception of the anti-SERCA2 pAb (raised in goat).

### 4.7. Statistical Analysis

Data were analyzed using Origin 8.0 SR6 (OriginLab Corporation, Northampton, MA, USA) and Microsoft Excel 2010 (Microsoft Corporation, Redmond, WA, USA) software. Values are reported as mean ± SEM (the number of experimental replicates (*n*) is indicated in the figures and/or figure legends). Statistical significance was evaluated by unpaired two-tailed Student’s t-test, a *p*-value < 0.05 being considered statistically significant (* *p* < 0.05, ** *p* < 0.01, *** *p* < 0.001).

## Figures and Tables

**Figure 1 ijms-20-05151-f001:**
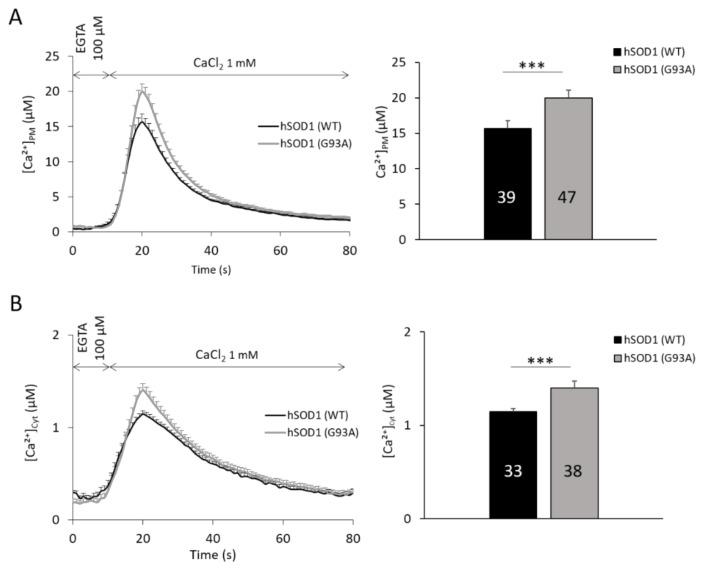
hSOD1(G93A) primary spinal cord astrocytes display enhanced store-operated Ca^2+^ entry (SOCE) compared to controls. hSOD1(WT) (black) or hSOD1(G93A) (grey) primary spinal astrocytes were transduced with lentiviral vectors encoding AEQ_PM_ or AEQ_Cyt_ targeted to sub-PM microdomains or bulk cytosol, respectively, and [Ca^2+^]_PM_ (**A**) and [Ca^2+^]_Cyt_ (**B**) fluctuations were recorded after SOCE stimulation. To maximally activate SOCE, we firstly depleted intracellular Ca^2+^ stores by incubating astrocytes in a Ca^2+^-free buffer containing EGTA (100 μM), and then perfused cells with a CaCl_2_ (1 mM)-containing solution, as indicated by arrows (see Materials and Methods for further details on the stimulation protocol). The average kinetics of the recorded local Ca^2+^ transients (left panels), and the bar diagrams (right panels) reporting the corresponding [Ca^2+^] peak values, clearly indicate that hSOD1(G93A) astrocytes have significantly higher Ca^2+^ influx in both cell compartments following SOCE than hSOD1(WT)-expressing cells. Here and after, reported values are mean ± SEM, numbers inside bars indicating the number of replicates (*n*), i.e., total coverslips analyzed from at least 6 different cell cultures for each experimental condition. Peak values are in sub-PM microdomains, 15.65 ± 1.11 μM in hSOD1(WT), 19.97 ± 1.10 μM in hSOD1(G93A); in the cytosol, 1.15 ± 0.03 μM in hSOD1(WT), 1.40 ± 0.07 in hSOD1(G93A). *** *p* < 0.001, unpaired two-tailed Student’s *t*-test.

**Figure 2 ijms-20-05151-f002:**
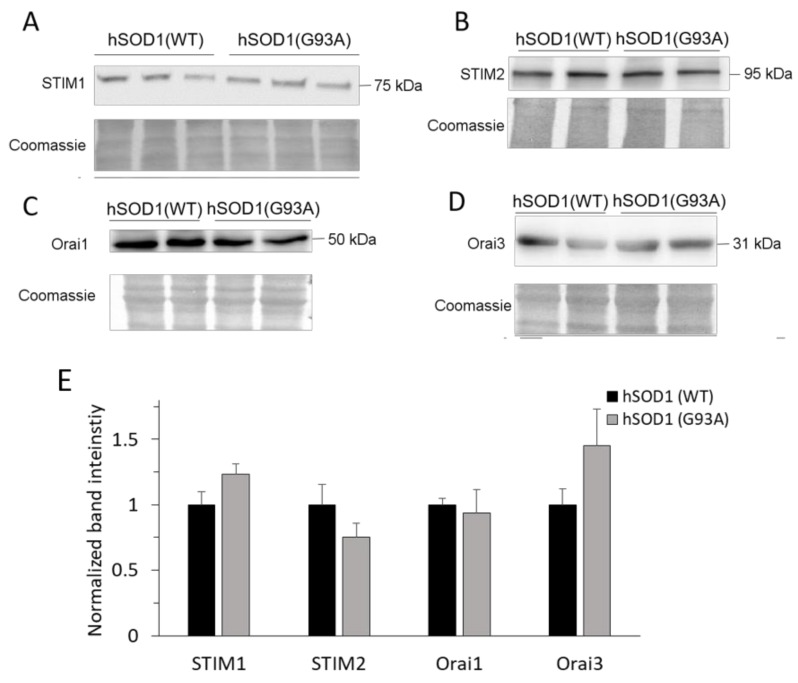
SOCE-regulating STIM and ORAI proteins are similarly expressed in primary astrocytes with the two hSOD1 genotypes. Proteins (18 μg for each sample) from astrocyte lysates were resolved by SDS-PAGE, electroblotted onto polyvinylidene difluoride (PVDF) membranes, and analyzed by Western blot (WB) for the expression of proteins belonging to the SOCE machinery, i.e., STIM1 (**A**), STIM2 (**B**), Orai1 (**C**) and Orai3 (**D**), using suitable antibodies. In each panel, the upper figures report a representative WB of the protein of interest (run at least in duplicate), and the lower figures show the corresponding Coomassie blue-stained membrane. Molecular weights are reported on the right of each WB panel. The bar diagram of panel (**E**) reports the densitometric analysis of immunoreactive bands for the different proteins normalized to the optical density of the corresponding Coomassie blue-stained lane. Densitometric data are reported as fraction of the mean value obtained with hSOD1(WT) cells. No significant difference (unpaired two-tailed Student’s *t*-test) exists for any of the analyzed proteins between hSOD1(WT) and hSOD1(G93A) astrocytes. *n* = 6 different biological replicates (i.e., different primary cultures) for each hSOD1 genotype and each target protein. Full-size images of WBs are reported in Appendix A.

**Figure 3 ijms-20-05151-f003:**
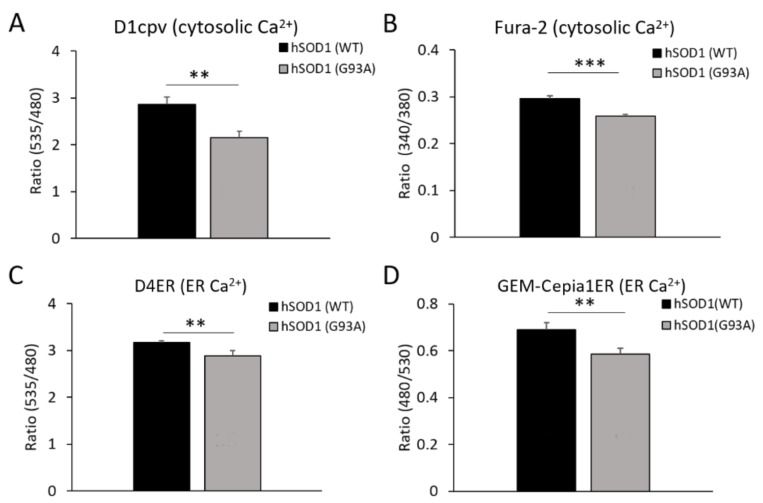
hSOD1(G93A) astrocytes have reduced basal Ca^2+^ levels in the cytosol and the ER lumen compared to the healthy counterpart. For measuring the basal [Ca^2+^] in the cytosol, primary spinal astrocytes were transfected with a plasmidic vector encoding the cameleon genetically-encoded Ca^2+^ indicators (GECI) D1cpt (**A**) or loaded with the chemical Ca^2+^ indicator Fura-2 (**B**). Both the fluorescence resonance energy transfer (FRET) signal (i.e., the fluorescence ratio between the FRET-acceptor yellow fluorescent protein (YFP) (535 nm) and the FRET-donor CFP (480 nm)) of the cameleon and the fluorescence ratio between the 340 nm and 380 nm excitation wavelengths of Fura-2 underscore significantly reduced cytosolic basal Ca^2+^ levels in hSOD1(G93A) astrocytes compared to the hSOD1(WT) counterpart. For measuring the basal [Ca^2+^] in the ER lumen, astrocytes were transfected with plasmids coding for the ER-targeted GECIs D4ER cameleon (**C**) or GEM-Cepia1ER (**D**). Both the FRET signal (D4ER) and the fluorescence ratio between the 480 nm and 530 nm excitation wavelengths (GEM-Cepia1ER) indicate that the basal ER [Ca^2+^] is significantly lower in hSOD1(G93A) astrocytes compared to healthy cells. Reported data were collected in at least 12 coverslips from at least 4 different primary cultures for each experimental condition. ** *p* < 0.01; *** *p* < 0.001, unpaired two-tailed Student’s *t*-test.

**Figure 4 ijms-20-05151-f004:**
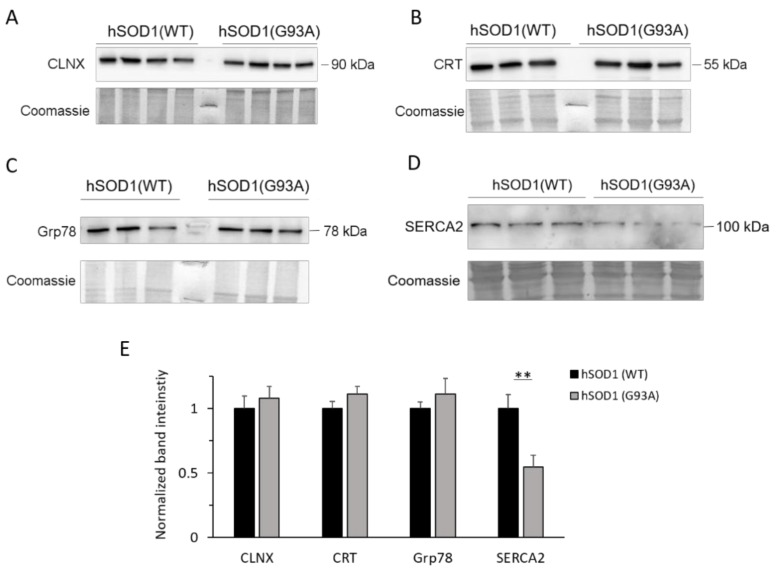
Familial amyotrophic lateral sclerosis (fALS) astrocytes express comparable levels of ER stress-related proteins but lower amounts of SERCA compared to the healthy counterpart. Protein extracts from astrocytes were analyzed by WB for the expression of the Ca^2+^-dependent ER stress markers calnexin (CLNX, **A**) and calreticulin (CRT, **B**), the unfolded protein response regulator Grp78 (**C**), and the sarco-endoplasmic reticulum Ca^2+^-ATPases (SERCA, **D**). In each panel, the upper figures report a representative WB of the protein of interest (run at least in triplicate), and the lower figures show the corresponding Coomassie blue-stained membrane. The bar diagram of (**E**) reports the densitometric analysis of immunoreactive bands for the different proteins normalized to the optical density of the corresponding Coomassie blue-stained lane. While no difference is observed in the expression of CLNX, CRT, and Grp78, SERCA2 is significantly downregulated in hSOD1(G93A) astrocytes compared to the hSOD1(WT) counterpart. *n* = 8 (SERCA), 6 (other target proteins) different primary cultures for each hSOD1 genotype; ** *p* < 0.01, unpaired two-tailed Student’s t-test. Other experimental details are as in the legend to Figure 2. Full-size images of WBs are reported in Appendix A.

**Figure 5 ijms-20-05151-f005:**
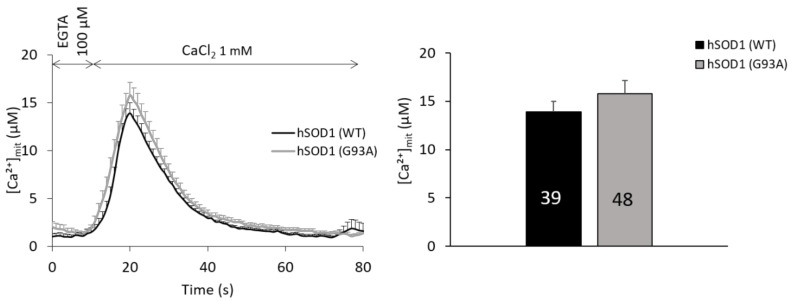
No difference is observed in SOCE-evoked mitochondrial Ca^2+^ transients in primary astrocytes expressing WT or mutant (G93A) hSOD1. Mitochondrial Ca^2+^ transients induced by SOCE activation were measured in hSOD1(WT) or hSOD1(G93A) astrocytes by means of lentivirally-delivered AEQ_mit_ targeted to the mitochondrial matrix. The average traces reported on the right and the corresponding peak values reported in the bar diagram on the left provide comparable results for the two astrocyte populations (unpaired two-tailed Student’s t-test). Peak values are: 13.88 ± 1.09 μM in hSOD1(WT), 15.76 ± 1.38 μM in hSOD1(G93A). Other experimental details are as in the legend to Figure 1.

**Figure 6 ijms-20-05151-f006:**
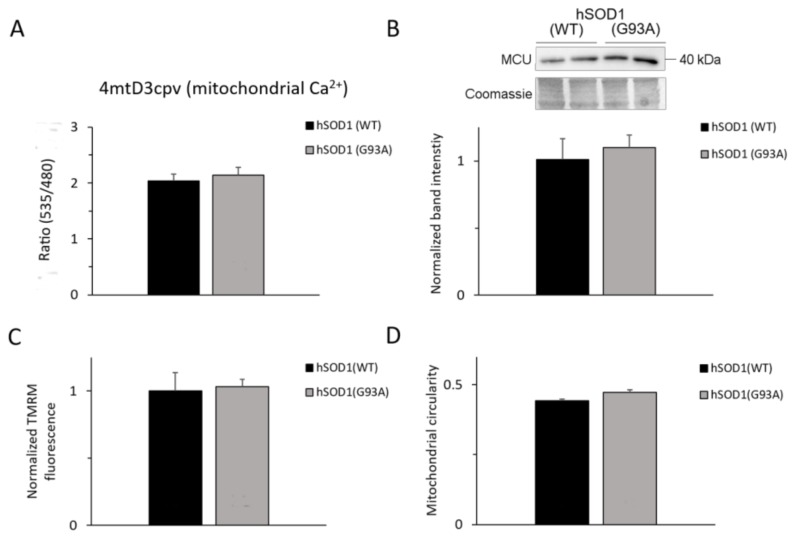
Mitochondrial Ca^2+^-related and morphologic parameters are unaffected by the expression of hSOD1(G93A). For measuring the basal Ca^2+^ level in the mitochondrial matrix (**A**), astrocytes were transfected with a plasmid encoding the mitochondrially-targeted cameleon probe 4mtD3cpv, and the FRET signal was measured 72 h after transfection. The expression level of the ion transporting subunit of the mitochondrial Ca^2+^ uniporter (MCU) complex (**B**) was assessed by WB. The upper part of the panel shows a representative WB from two different astrocyte lysates for each hSOD1 genotype and the corresponding Coomassie blue staining, while the bar diagram in the lower part of the panel reports the densitometric analysis of MCU-immunoreactive bands (as described in Materials and Methods and the legend to Figure 2). A full-size image of the WB is reported in Appendix A. For measuring Δψ_m_ (**C**), astrocytes were loaded with the mitochondrial potentiometric tetramethyl-rhodamine methyl ester (TMRM) probe and the TMRM signal in mitochondria-rich regions was measured by means of a suited fluorescence microscope (for further details, see Materials and Methods). Reported data are normalized to the mean value of hSOD1(WT) astrocytes. For estimating mitochondrial morphology (**D**), astrocytes were transfected with a plasmid encoding a red-fluorescent protein targeted to mitochondria. 72 h after transfection, fluorescence images were collected by a confocal microscope, and mitochondrial circularity was calculated as described in Materials and Methods. None of the four measured mitochondrial parameters significantly differ between hSOD1(WT) and hSOD1(G93A) astrocytes (unpaired two-tailed Student’s *t*-test). Reported data are from at least 12 coverslips from at least 4 different primary cultures (panels A, C and D) or from 6 different primary cultures (panel B) for each hSOD1 genotype.

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
