# Peer review of "ALS-Associated SOD1(G93A) Decreases SERCA Pump Levels and Increases Store-Operated Ca2+ Entry in Primary Spinal Cord Astrocytes from a Transgenic Mouse Model"

_ijms, 2019, doi:10.3390/ijms20205151_

Round 1
Reviewer 1 Report
Norante et al. analyzed subcellular Ca2+dynamics in cultured astrocytes isolated from new-born hSOD1(G93A) mice using genetically encoded Ca2+indicators. The authors found that Ca2+level in cytosol and ER lumen, but not mitochondria, decreased in hSOD(G93A) astrocytes. Among Ca2+handling proteins the authors have analyzed, SERCA2 protein level was decreased. The authors conclude the decrease in Ca2+uptake into ER through SERCA2 is a main cause of the abnormality of Ca2+regulation in hSOD1(G93A) astrocytes. This is interesting manuscript which provide information how Ca2+handling mechanism is disrupted in astrocytes in neurodegenerative disease models. This includes informative observations to the field. However, there are some concerns which need to be addressed.
First, the authors depleted ER calcium using EGTA to image Ca2+through store-operated Ca2+entry (SOCE) in Figure 1 and found that larger Ca2+signals by the SOCE protocol. In this experiments, SERCA2 activity is still active because SERCA2 pump was not inhibited. Therefore, the larger Ca2+signals induced by the protocol could be simply due to less Ca2+uptake through reduction of SERCA2, not increase Ca2+entry. Ca2+uptake should be inhibited in this experiments. Similarly, SERCA2 should be inhibited in Figure 5 experiments to make the interpretation clearer.
Second, the authors clearly show lower Ca2+level in ER of hSOD1(G93A) astrocytes using two different genetically encoded Ca2+indicators which are targeted to ER. To confirm this observation further, it is nice to see the effect of thapsigargin. The authors would expect smaller Ca2+rise by thapsigargin. This may provide information why the discrepancy occur between present study and past study by Kawamata et al., 2014.
Third, since astrocytes express a plethora Gq-protein coupled receptors to induce Ca2+signals. Ca2+signals induced by activation of GqPCR at maximal concentration of agonist should be smaller, if Ca2+uptake via SERCA is impaired in hSOD1(G93A) astrocytes. This could be an important observation which make the authors argument stronger.
Finally, there is no image for Ca2+imaging. Therefore it is difficult to assess how Ca2+measurement works. It is nice to show representative images for those experiments. Also, it is nice to show images for TMRM (mitochondrial membrane potential).
Reviewer 2 Report
The Authors studied here calcium fluxes induced by SOCE activation in cultured astrocytes from newborn wtSOD1- and SOD1(G93A)-expressing mice. To this aim, they applied quite sophisticated techniques using a battery of calcium dyes genetically-encoded to measure calcium oscillations in well-defined cell domains. They report a decrease of the cytosolic and ER calcium content, the latter possibly linked to reduced expression of SERCA, and an increase of SOCE-mediated calcium refilling, triggered by depauperating the ER calcium content.
The SOCE increase in SOD1(G93A) mouse astrocyte is quite limited and most of the other readouts were found unmodified. One may wonder which is the impact of this event on the disease. Possibly, the results may have been influenced by the use of newborn astrocytes. SOD1(G93A) mice display very limited motor neuron death, astrocyte activation and inflammation at this age and the animals do not show any clinical symptom of the disease. The use of astrocytes from adult, symptomatic mice (i.e. matured in a pathological environment) could disclose more relevant effects. This part of the research would have improved considerably the impact of the work. Conversely, the modification observed, although limited, are very precocious during disease progression, supporting their etiological role. The Authors already discussed this item (page 13, lines 7-15). They should also take into consideration the above arguments in the discussion.
A number of major and minor point s should be addressed by the authors to improve the impact of their work.
Major points
1. SOCE stimulation.
The Authors exposed astrocytes to a calcium-free + EGTA medium for 1 hour to deplete the ER and induce SOCE. This is in my experience a quite unusual procedure that certainly produces long-lasting stress to cell biochemistry and membranes. Why other methods, such as caffeine or ionomycin, to deplete the stores, or thapsigargin, to block SERCA in the ER, have not been considered?
Did the Authors control that this procedure really emptied the ER stores? Looking at fig 1, it seems that a residual calcium concentration (about 0.5 µM) is still present in PM and cytoplasm, even after such a drastic procedure. Moreover, the resting calcium concentration in the cytoplasm does not vary in calcium-free medium and after calcium replenishment. What about ER?
2. Calcium measures.
Figures 1 and 5 are unclear. The Authors exposed astrocytes to calcium-free + EGTA environment for 1 hour and apparently 20 s of this time has been displayed in the figures. If this were the case, however, the switch to the calcium-containing medium seems to occur when calcium already reached the peak of concentration.
Only Figure 1 and 5 show the true concentrations of calcium. In the other circumstances (Figs. 3 and 6A) the fluorescence ratios are reported. This procedure is acceptable to show calcium fluctuation in a specific preparation but it inaccurate when two different samples are compared. For instance, to calibrate calcium concentration using the Fura-2 dye, Rmax, Rmin, background-340 and background-380 are necessary to be considered, besides the two arbitrary fluorescence values at 340 and 380 nm.
3.SERCA and calcium balance in ER and cytoplasm.
The Authors found reduced levels of calcium both in the ER and cytoplasm of SOD1(G93A) astrocytes and a reduction of SERCA expression was indicated as a possible cause. If this were the case, I would have expected an increase in cytoplasmic calcium since the two phenomena should be the opposite. A reduced entry of calcium into the ER should lead to increased values in the cytoplasm, also considering that mitochondria and plasma membrane calcium extruders do not seem to play a role in these oscillations.
4. Mitochondria involvement.
Indeed, the lack of modifications of mitochondria calcium oscillations is “quite surprising” or “paradox” since is a general belief that mitochondria and ER relate each other also exchanging calcium ions. Besides the very rapid time scale of calcium oscillations, suggested by the authors as a cause, the limited calcium level modifications may well play a role in the reported unmodified mitochondria functions and morphology.
In this context, I found inappropriate the debate of bioenergetic alterations just measuring the mitochondrial membrane potential but nor other parameters. Moreover, cell bioenergetic alterations not necessarily are linked to the mitochondrion: it is well known the role of glycolysis in the astrocyte bioenergetics. The Authors should discuss more in details this issue to justify their assumption.
FCCP was correctly used as a control in the membrane potential measurement to collapse the gradient. This control results should be described in the text or included as a supplementary figure.
5. Protein quantification.
In spite of the cited paper (ref. #106), normalization of band intensities using the Coomassie staining appears rather outdated. Generally speaking, this normalization is justified when the two or more sample to be compared does not share suitable housekeeping proteins. This is not the case of SOD1(G93A) and wtSOD1 astrocytes. Moreover, when the total protein intensity is utilized as a reference, the Ponceau Red staining would be more appropriate since, in this case, proteins are coloured in the same blotted membrane and not on the original gel. If the Authors would not decide to shift to an immunologically-detected reference protein and would continue to prefer their method, measures should be made and showed on the whole protein lane and not only in the portion related to the MW of the analyzed protein. Moreover, this choice should be justified.
The quality of the Coomassie images is poor everywhere.
6. S-glutathionylation.
The Authors detected an enhancement S-glutathionylation in SOD1(G93A) astrocytes but they do not clarify if and which protein(s) involved in SOCE function have been investigated. These not shown data should be disclosed to the reader. Subordinately, the relevant parts in the discussion should be omitted.
Minor points
The authors report five times that their results are “unprecedented” or described "for the first time”, which is quite obvious and should be smoothed.
There are too many references for a research paper (106). They are 51 only in the discussion (72 if we consider also the descriptive paragraphs at the beginning of each section of the Results). The number of references should be shrunk.
The first paragraphs of the Results sections are too itemized and should be reduced and/or moved elsewhere (Introduction/Discussion).
Results, page 8 lines 11-14: this correlation does not seem justified on the basis of the selected protein quantification only.
Sometimes, mutant SOD1-bearing astrocytes are reported in the text and in the legends to the figures as fALS cells. The SOD1(G93A) notation should be used everywhere.
Figure 2 and 3 legends report “Other experimental details are as in the legend to Figure 1”, that is incorrect.
Figure 3 and 4 legends: the statistical method used should be indicated (see legend to Fig. 1).
Figure 6 legend report that “data are normalized to the mean value of hSOD1(G93A)”. It should be corrected in hSOD1(WT).
Figure 2A, B, D: bands do not seem to be representative of the results reported in the bar graph, also in view that no obvious differences can be appreciated in the Coomassie normalization panels.
Figure 3: the notation “Basal [Ca2+]…” is misleading since only the fluorescence ratios are reported in the panels. Any notation should be placed on the top of each of the four panels.
Figure 6, panel A: it should be indicated that this is a calcium measure.
Discussion, page 12 line 4 from bottom: “This finding suggests that the ER of hSOD1(G93A) astrocytes has a lower Ca2+ buffering capacity”. Why? The Authors showed that the buffering proteins analyzed did not change (Fig. 4).
Methods, section 4.5: The Authors should describe in details how mitochondrial circularity can be calculated from the four parameters measured.
Reviewer 3 Report
Norante et al. investigated the change in calcium regulation in primary culture of astrocytes using calcium indicators in mutant SOD1 Tg mice. Previous work on primary astrocytic culture (Kawamata et al., 2014) has demonstrated that excess calcium release from the ER store upon stimulation and enhanced astrocytic store-operated calcium entry (SOCE) may underlie astrocyte-mediated neurotoxicity in mutant SOD1 Tg mice. The novelty of this work is demonstration of low basal calcium concentration in ER using various calcium indicators and a reduced expression of sarco-ER calcium-ATPase (SERCA). The authors did not fully discuss the discrepancy between the previous work or demonstrate whether their new finding has a role in neurotoxicity in mutant SOD1 Tg mice.
The authors used wild-type SOD1 transgenic mice as the control. To assess the effects of SOD1 transgene, use of non-Tg mice as the control are also necessary. In addition, copy numbers of the WT SOD1 and mutant SOD1 transgenes should be indicated.
Experiments were conducted on astrocytes at different culture time point. Culture time should be the same or time-dependent change may be necessary, particularly of STIM1 and SERCA. Ideally, to enable comparison with the previous study on Kawamata et al. (2014), results on primary culture of adult mouse-derived astrocytes will promote our understanding.
Kawamata et al. (2014) indicated an increase of STIM1 glutathionylation, but STIM1 did not differ between WT SOD1 and mutant SOD1 mouse astrocytes in this study. The authors should discuss the discrepancy of the results.
The authors claim that enhanced SOCE occurs in the absence of STIM1 downregulation in neonatal astrocytes and decreased Ca2+ buffering effects of ERs resulted from reduced expression of SERCA2. As ALS initiates in midlife, it is acceptable if a reduction of SERCA2 alone in neonatal astrocytes is not sufficient to induce disease, but combination of reduction in both STIM1 and SERCA2 in adult life initiate disease More information is necessary to conclude that SERCA2 downregulation also plays a role in aberrant Ca2+-buffering effects of ERs. 1) It is hard to evaluate the reduction of SERCA2 from the poor WB image in Fig. 4D. More informative image is required and confirmation by RT-PCR like in Fig. S2 for SERCA2b is required as well. 2) In addition, readers would like to know whether SERCA2 is downregulated in adult astrocytes. 3) Whether the downregulation of SERCA2 (or both STIM1 and SERCA2) contribute to reduced ER buffering function should be demonstrated. Authors can demonstrate by knockdown of SECA2 (alone or in combination with STIM1) in other cultured cells using the same calcium indicators. 4) To make this finding more meaningful, it is also necessary to demonstrate whether the culture medium of astrocytes knocked down for SERCA2 enhances neurotoxicity. 5) It is also interesting to know whether SERCA2 is downregulated in astrocytes in the spinal cord of mutant SOD1 Tg mice in situ.
Round 2
Reviewer 1 Report
No further objection.
Reviewer 2 Report
No comments
Reviewer 3 Report
The authors did not add any data required to improve their manuscript.
Although the authors claim that SOD1 Tg mice is a model of ALS, with the accumulation of evidence demonstrating the difference among SOD1 ALS, SOD1 Tg mice and sporadic ALS that accounts for the majority of ALS, SOD1 Tg mouse model has no longer been regarded as the mechanistic model of ALS. In addition, as ALS is an adult-onset motor neuron disease, findings on neonatal mice are remote from the ALS pathogenesis, particularly when these findings were not found in mature mice. What they can say is limited to the changes in neonatal mutant SOD1 Tg mice (not an ALS model) relative to wt SOD1 Tg mice. It is a leap of faith that the authors discuss pathogenic mechanism of ALS from what they found on the samples from neonatal mutant SOD1 Tg mice. The authors should delete all the discussion on ALS and restrict their discussion on how neonatal mutant SOD1 Tg mice are different from neonatal wt SOD1 Tg mice.